# High Incidence of Strawberry Polerovirus 1 in the Czech Republic and Its Vectors, Genetic Variability and Recombination

**DOI:** 10.3390/v13122487

**Published:** 2021-12-11

**Authors:** Jana Fránová, Ondřej Lenz, Jaroslava Přibylová, Radek Čmejla, Lucie Valentová, Igor Koloniuk

**Affiliations:** 1Department of Plant Virology, Institute of Plant Molecular Biology, Biology Centre, Czech Academy of Sciences, 370 05 České Budějovice, Czech Republic; lenz@umbr.cas.cz (O.L.); pribyl@umbr.cas.cz (J.P.); 2Laboratory for Molecular Biology, Research and Breeding Institute of Pomology Holovousy Ltd., Holovousy 129, 508 01 Hořice, Czech Republic; radek.cmejla@vsuo.cz (R.Č.); lucie.valentova@vsuo.cz (L.V.)

**Keywords:** strawberry, mixed virus infection, RT-PCR, virus transmission

## Abstract

In total, 332 strawberry plants from 33 different locations in the Czech Republic with or without disease symptoms were screened by RT-PCR for the presence of strawberry polerovirus 1 (SPV1) and five other viruses: strawberry mottle virus, strawberry crinkle virus, strawberry mild yellow edge virus, strawberry vein banding virus, and strawberry virus 1. SPV1 was detected in 115 tested strawberry plants (35%), including 89 mixed infections. No correlation between symptoms and the detected viruses was found. To identify potential invertebrate SPV1 vectors, strawberry-associated invertebrate species were screened by RT-PCR, and the virus was found in the aphids *Aphis forbesi*, *A. gossypii*, *A. ruborum*, *A.*
*sanquisorbae*, *Aulacorthum solani*, *Chaetosiphon fragaefolii*, *Myzus ascalonicus*, and several other non-aphid invertebrate species. SPV1 was also detected in aphid honeydew. Subsequent tests of *C. fragaefolii* and *A.*
*gossypii* virus transmission ability showed that at least 4 h of acquisition time were needed to acquire the virus. However, 1 day was sufficient for inoculation using *C. fragaefolii*. In conclusion, being aphid-transmitted like other tested viruses SPV1 was nevertheless the most frequently detected agent. Czech SPV1 isolates belonged to at least two phylogenetic clusters. The sequence analysis also indicated that recombination events influence evolution of SPV1 genomes.

## 1. Introduction

More than 30 viral species have been described in strawberries in the past [1], and the number of newly identified viruses is increasing [2,3,4,5,6,7,8,9]. The most economically important and widespread of these viruses are strawberry mottle virus (SMoV), strawberry crinkle virus (SCV), strawberry mild yellow edge virus (SMYEV) and strawberry vein banding virus (SVBV), especially when they occur in mixed infections [1,2,10,11,12].

In 2015, a novel virus—strawberry polerovirus 1 (SPV1)—was reported in complex viral mixtures in strawberries affected by disease causing acute declines in Canada [4]. Subsequently, cases of SPV1 were reported in the USA [12,13], Argentina [14], and the Czech Republic [7]. SPV1 belongs to the family *Solemoviridae*, which includes icosahedral plant viruses with small positive-sense RNA genomes approximately 5 kb in length encoding 4–5 proteins. In addition to vegetative propagation, grafting, and mechanical transmission, some solemoviruses are transmitted by insect vectors [15]. However, the vectors of SPV1 are not yet known.

Therefore, the aim of this study was an epidemiological survey of SPV1 in strawberry plants in the Czech Republic to screen its incidence, to identify its putative vectors (including experimental vector-mediated plant-to-plant transmission), and to assess genetic variability and phylogeny inference of circulating strains. As mixed infections are frequent in strawberries, the other most important and widespread strawberry viruses in the region, SMoV (family: *Secoviridae,* genus: *Stramovirus*), SCV (family: *Rhabdoviridae*, genus: *Cytorhabdovirus*), SMYEV (family: *Alphaflexiviridae*, genus: *Potexvirus*), as well as SVBV (family: *Caulimoviridae*, genus: *Caulimovirus*), and recently identified strawberry virus 1 (StrV-1, family: *Rhabdoviridae*, genus: *Cytorhabdovirus*), were also included in the screening.

## 2. Materials and Method

### 2.1. Plant Samples

During 2016–2021, a total of 332 strawberry plants (296 *Fragaria x ananassa* Duchesne, 16 *F. vesca semperflorens* cv. Rujana, 20 woodland *F. vesca*) were sampled from 25 production farms, one nursery plantation, four private gardens and three forests in eight regions in the Czech Republic (Appendix A, Appendix A). For high-throughput sequencing (HTS), two symptomatic plants of *F. ananassa* (isolates 34/2016 and 138/2020), and one plant of *F. vesca* cv. Rujana (isolate 1/2017) showing perspicuous virus-like symptoms [7] were selected. Samples were preferably collected from 2- to 3-year-old strawberry fields.

Leaves of weeds and fruit trees growing near or among the strawberry plants were collected as potential virus reservoirs.

Both *F. ananassa* Duch. cv. Čačanská raná plant and *F. vesca* ‘Alpine’ seeds were kindly provided by Mrs. M. Erbenová from the Research and Breeding Institute of Pomology Holovousy in 1993. *F. vesca* ‘Alpine’ (‘Alpine’ below) plants grown from seeds, daughter plants of *F. ananassa* cv. Čačanská raná (ČRM3) grown from runners and *F. vesca* ‘Alpine’ (No. 814) material graft inoculated from *F. ananassa* ČRM3 were used to test virus transmission by aphids.

### 2.2. Identification of Invertebrates and Honeydew Collection

Invertebrates were collected from SPV1-positive strawberry plants or trapped in strawberry fields and gardens. Their presumptive identity was determined by the visual examination of morphological characteristics. Total RNA isolated from single aphids or groups (ranging from two to ten individuals) of aphids, ants, enchytraeids and two leafhoppers was forwarded for molecular identification. The primer pair LCO1490/HCO2198 [16] was used to amplify a 658 bp fragment of cytochrome C oxidase subunit 1 (COI) mRNA. The PCR products were either directly subjected to Sanger sequencing or were cloned into the pGEM T-Easy vector (Promega, Road Madison, WI, USA), and plasmid DNAs from selected clones were sequenced using vector-specific primers (Eurofins Genomics, Ebersberg, Germany).

Empty Petri dishes were placed under the leaves of *F. vesca* cv. Rujana (later Rujana; 1/2017) and *F. ananassa* ČRM3 occupied by *Aphis ruborum* (*A. ruborum*) (Börner & Schilder, 1931) and *Chaetosiphon fragaefolii* (*C. fragaefolii*) (Cockerell, 1901). Honeydew was trapped in the dishes overnight, dissolved in RPL extraction buffer (350 µL) from a Ribospin Plant purification kit (GeneAll, Seoul, Korea) and used for total RNA extraction, SPV1 amplification and Sanger sequencing.

### 2.3. RNA Extraction and cDNA Synthesis, Reverse Transcription Polymerase Chain Reaction (RT-PCR), and Sanger Sequencing

Total RNA was extracted from 50 mg of the fresh leaf blades of strawberries, weed plants, fruit trees, and from the whole body (aphids and small invertebrates) or head and thorax of insects using either a Gene JET Plant RNA Purification Kit (Thermo Fisher Scientific, Vilnius, Lithuania) or a Ribospin Plant (GeneAll) following the manufacturers’ protocols. The quality and quantity of RNA were measured on a Nanodrop 1000 spectrophotometer (Thermo Fisher Scientific). The obtained RNA was reverse-transcribed using the M-MLV Reverse Transcriptase kit (Invitrogen, Carlsbad, CA, USA) according to the manufacturer’s recommendation.

The RT-PCR amplification of the mitochondrial NADH dehydrogenase nad5 mRNA using the primers Atropa Nad2.1a/2b was performed as an internal amplification control for plant samples and a diet control in invertebrates [17]. Only positive plant samples were subsequently used for virus detection.

For two-step RT-PCR, 1 µL of a cDNA preparation was added to a mixture of 10 µL of 2× PPP Master Mix (Top-Bio, Vestec, Czech Republic), 8 µL of PCR-grade H_2_O, and 0.5 µL of each primer (0.2 µM). All primers used in the study, their sequences and the corresponding amplification conditions are listed in Appendix A. Reaction mixtures devoid of cDNA templates served as no-template controls. Each PCR product (4 µL) was analyzed by electrophoresis in a 1% agarose gel pre-stained with GelRed (Biotium, Hayward, CA, USA). Data on infection counts were analyzed using R software [18] and UpSetR package [19].

The PCR products (16 µL, from plants as well as from invertebrates) were excised from 1.5% agarose gel and purified using the Expand Combo mini kit (GeneAll). The products were Sanger sequenced from both directions (Eurofins Genomics, Luxembourg).

### 2.4. Detection and Quantification of SPV1 in Individual Aphids by RT-qPCR

*Aphis gossypii* (*A. g**ossypii*) (Glover, 1877), *Aphis sanguisorbae* (*A. s**anguisorbae*) (Schrank, 1801), and *C. fragaefolii* adults were collected from native colonies feeding on *F. ananassa* ČRM3 plants that were positive for SPV1, SMoV, SCV, and StrV-1. Total RNA was isolated from individual specimens using TRI reagent (Merck, Kenilworth, NJ, USA). The extracted RNA was quantified using a Nanodrop spectrophotometer and the Qubit HS RNA assay (Thermo Fisher Scientific). cDNA was generated using the Maxima First Strand cDNA Synthesis Kit for RT-qPCR followed by dsDNase treatment (Thermo Fisher Scientific). RT-qPCR was performed with a CFX96 real-time PCR detection system (Bio-Rad, Hercules, CA, USA) and 5× HOT FIREPol EvaGreen qPCR Mix Plus (Solis BioDyne, Taru, Estonia) reaction mixture. Two endogenous controls, mitochondrial 16S ribosomal RNA and succinate dehydrogenase B mRNA, were employed for the normalization of expression levels (Appendix A). Each run included positive, negative, no-reverse-transcriptase and no-template controls. All reactions were performed in triplicate using eight biological replicates. The data were analyzed using Bio-Rad CFX Maestro 1.1 (Bio-Rad) and R software [18].

### 2.5. Aphid Transmission SPV1 to F. vesca ‘Alpine’ Plants

All experiments with aphids were conducted in custom-built mesh cages in an air-conditioned greenhouse under a 16 h light/8 h dark cycle and temperature at 24 °C.

#### 2.5.1. Transmission Experiment with Various Acquisition Access and Inoculation Access Periods

Virus-free colonies of *A. gossypii* and *C. fragaefolii* were established as individual lines from single newborn 1st-instar aphid and were cultured on seed-grown ‘Alpine’ plants at 18 °C. Detached leaves from ‘Alpine’ (No. 814) plant simultaneously infected with SPV1, SMoV and SCV were used as SPV1 source material at the beginning of the study due to the absence of any experimental plants infected with SPV1 alone (Appendix A).

For SPV1 transmission assays, 256 individual *A. gossypii* wingless adults were divided into 16 groups (each of 16 individuals) with varying acquisition access periods (AAPs) and inoculation access periods (IAPs) of 10 min, 4 h, 24 h, or 48 h. Following feeding on the leaves of ‘Alpine’ No. 814 plant, the aphids were transferred to 64 seed-grown ‘Alpine’ plants (four aphids per plant). Altogether, 16 variants of the transmission trials were conducted, and each experiment was performed in four replicates at 24 °C (Figure 1).

Each individual group of aphids (*n* = 64) collected from ‘Alpine’ plants after IAP was tested for the presence of SPV1. After IAP, the ‘Alpine’ plants (*n* = 64) were sprayed with FAST M (active ingredient: deltamethrin 0.12 g/L) and were examined for the presence of SPV1, SMoV, and SCV at 40 days post inoculation (dpi). The plants were monitored for viral disease symptoms daily for 4 months.

Additionally, *C. fragaefolii* were transmitted to leaf of the ‘Alpine´ plant (infected with SPV1 solely; the plant was obtained independently during the study) for AAP of 10 min and 4 h. After AAP and 1 h of fasting period, four batches of aphids (each with 4 aphid individuals for each APP) were RT-PCR tested for SPV1 presence.

#### 2.5.2. Transmission Experiment with Not-Limited AAP and Limited IAP

Individual adults of *A. gossypii* (*n* = 40) and *A. sanguisorbae* (*n* = 20) were transferred from native colonies feeding on SPV1-positive *F. ananassa* ČRM3 plants to 60 plants of *F. vesca* ‘Alpine’. After IAP (24 h for *A. gossypii* (*n* = 40) and *A. sanguisorbae* (*n* = 10) and 48 h for *A. sanguisorbae* (*n* =10)), each individual aphid was analyzed by RT-qPCR for the presence of SPV1 using the primers SPV-12f/2r (Appendix A). The amplification of COI mRNA [16] was performed as an aphid endogenous control. The recipient plants (*n* = 60) were processed similarly at 60 dpi.

#### 2.5.3. Transmission Experiment with Not-Limited AAPs and IAPs

Eight individuals of *C. f**ragaefolii* collected from a Rujana 7/2017 plant (positive for SPV1, SMoV, SCV, SMYEV, and StrV-1) were transferred to ‘Alpine’ plant. After one month of cultivation at 24 °C, 10 aphids were tested for the presence of SPV1 and other viruses. Plant was sprayed with FAST M and, after 10 days, was examined by RT-PCR for the presence of the viruses.

Similarly, batches of *C. f**ragaefolii* aphids were transmitted from *F. vesca* ‘Alpine’ (infected with SPV1 solely) to *F. vesca* ‘Alpine’ plants for IAPs of 10 min, 4 h, 8 h, 1, 2, 3, 5, 7, 11, 14, and 17 days. Following the IAP, plants were sprayed with FAST M. After one month, the plants were examined by RT-PCR for the presence of viruses and were continuously observed for disease symptoms.

### 2.6. HTS and Sequence Analyses

Sequencing libraries were prepared using the Collibri Stranded RNA Library Prep Kit for Illumina (Thermo Scientific) from total RNA previously depleted of ribosomal RNA with a RiboMinus kit (Thermo Scientific) following the manufacturer’s recommendations. After quantification and quality control, the libraries were processed using NovaSeq6000. The obtained 150 bp paired-end reads were quality and adapter trimmed and analyzed with CLC Genomics Workbench 9.5.1 (Qiagen, Hilden, Germany). Briefly, the trimmed reads were de novo assembled with minimum contig size of 450 bp. The resulting sequences were compared against local database of custom viral proteins using BLASTx (E-value cutoff 1e-5) in Geneious Prime 2021.1.1 (Biomatters Ltd., Auckland, New Zealand). The potential viral hits were then compared against GenBank nr database (20 October 2021; e-value cutoff 1e-3). Sequence alignments and phylogenetic trees were produced using Geneious 9.1.8 (Biomatters Ltd., Auckland, New Zealand) and Geneious-integrated tools (ClustalW, MAFFT, GeneiousTree builder (Jukes–Cantor distance model, Neigbour-joining method, bootstrap with 1000 replicates)). Recombination analyses of the obtained alignments were performed with RDP5 software (RDP, v. 5.05 Beta) with default settings and window size = 50 [20]. Synonymous and nonsynonymous mutations were manually counted from the aligned nucleotide sequences, with translations displayed in the Geneious software.

## 3. Results

### 3.1. Symptoms and Virus(es) Presence

The visual inspection of strawberry farms, nurseries, private gardens, and woodland strawberries mostly revealed a low incidence of symptomatic plants. Therefore, plants with virus-like symptoms (251 *F. ananassa*, 12 *F. vesca semperflorens*, and 6 forest *F. vesca* plants) were preferentially sampled over asymptomatic strawberries (45 *F. ananassa*, 4 *F*. *vesca semperflorens*, and 14 forest *F. vesca* plants). Symptoms observed on the strawberry plants ranged from chlorosis, mosaic, irregular vein clearing and necrosis, and the reddening and deformation of leaves to stunting and whole-plant decline (Figure 2 and Figure 3). The most common symptoms were dwarfism and chlorosis. An overview of the strawberry plants, their symptoms, and the detection of SPV1, SMoV, SCV, SMYEV, SVBV, and StrV-1 is shown in Appendix A.

Up to 80% of plants were observed to be symptomatic among older plantings of *F. ananassa* cv. Elkat on two strawberry farms in the Moravian-Silesian Region (locality MS-1-F, MS-2-F). The infected plants often showed irregular vein clearing and necrosis and/or mosaic (Figure 2). Viruses were detected in both symptomatic and symptomless plants.

Prominent leaf reddening and general weakening of plant growth were observed among *F. ananassa* plants at a farm in the Pilsen Region in 2020 (locality P-1-F, Figure 3A). SPV1, SMoV, and SCV were previously detected on this farm in 2017. In 2020, all 31 examined plants were infected with different combinations of SMoV, SCV, SMYEV, and SPV1. Neither SVBV nor the recently described cytorhabdovirus StrV-1 were found at that location.

At the Z-1-F locality in the Zlín Region (Figure 3B), severe preliminary leaf reddening, irregular shapes, chlorosis of young leaves, and declines of plants were observed; SMoV (*n* = 2), SCV (*n* = 1), and SMYEV (*n* = 1) were found at this site sporadically among 22 examined plants. Twelve plants showing decline and dieback symptoms were RT-PCR negative for the tested viruses. We also detected the presence of the fungal agents *Alternaria* sp., *Coniella fragariae*, *Fusarium* sp., *Pythium sylvaticum*, *Rhizoctonia* sp., and *Verticilium* sp. (data not shown). Similar symptoms of leaf reddening and death of plants were found on another farm (South Moravian Region, locality SM-5-F) and in the Olomouc Region (locality O-1-F) (Appendix A).

There were plants lacking any symptoms of viral disease at some sites. At one location (Zlín Region, locality Z-2-F), only a slight stunting of plants was observed, and the viruses were not detected at that site.

Among 45 examined symptomless *F. ananassa* plants, 29 plants (64%) were tested negative for virus presence. However, the remaining plants were virus-positive, with either single (SPV1, SCV, or SMoV) or mixed infections (different combinations of SPV1, SMoV, SCV, SMYEV, and StrV-1; Appendix A).

For *F. ananassa* samples, a chi-square test of independence showed that there was no significant association between one of the tested viruses and symptomatic phenotype (X2 (5, *n* = 1775) = 0.0005, *p* = 0.99), thus excluding association of any particular virus and observed symptoms. Then, comparison of symptomatic samples showed strong association with virus presence (X2 (1, *n* = 1775) = 87.63, *p* = 7.9 × 10^−21^) regardless of its nature (either viral species or single/mixed infection), meaning there was a significant relationship between virus infection and observed symptoms.

Sixty plants of *F. vesca* cv. Rujana were grown from seeds purchased in a shop and had been cultivated in a private garden in Třísov, South Bohemia, (locality SB-10-G) since spring 2016. During 2016, increasingly severe virus-like symptoms were observed in a growing number of plants. By April 2017, all plants already had symptoms of mosaic, leaf and flower malformation, and dwarfing, and some plants were declining. All randomly selected plants (*n* = 11) were positive for SPV1 in combination with SMoV, SCV, StrV-1, and SMYEV. At the end of 2017, all plants were removed from the garden. In 2018, 30 seed-grown strawberries were planted approximately 50 m from the original garden. Following disease symptom appearance, five plants were tested in September 2019. RT-PCR showed the presence of StrV-1 in a plant with dwarfism and mosaic symptoms and SPV1 in combination with SCV and StrV-1 in one symptomless plant. Colonies of *C. f**ragaefolii*, *A. ruborum*, *A. sanguisorbae*, and *Aulacorthum solani* (*A. solani*) (Kaltenbach, 1843) aphids were notably found on the plants in both 2017 and 2019.

### 3.2. Incidence of SPV1, Frequency of Mixed Infections

Among the tested viruses, SPV1 was the most frequently found (*n* = 115; 35%), followed by SMoV (*n* = 100; 30%), SMYEV (*n* = 96; 29%), SCV (*n* = 88; 27%), and StrV-1 (*n* = 70; 21%) (Figure 4). In *F. ananassa* plants, RT-PCR revealed SPV1 either alone (*n* = 26) or in coinfections with other tested viruses (*n* = 77). All 12 SPV1-positive *F. vesca* cv. Rujana garden plants were coinfected with either two or three other viruses (SMoV, SCV, and StrV-1). An isolate of SVBV (67/2019) was identified in only one plant of *F. ananassa* cv. Faith (locality MS-2-F), and its complete sequence was deposited in GenBank under Acc. No. MW387997. The only viruses found to infect *F. vesca* in the forest were SMoV and StrV-1 (locality SB-8-W and SB-12-W), while SMoV, SCV, and StrV-1 were detected in wild-grown *F. vesca* as well as cultivated *F. ananassa* in a garden (locality SB-9-G, Appendix A).

The production fields with the highest virus incidences were located in two strawberry farms (localities P-1-F and MS-2-F) in the Pilsen and Moravian-Silesian regions (distance, approximately 450 km; only farms where more than five plants were examined were included in the analysis). SPV1 was repeatedly identified in samples collected at these sites during 2017–2020. The tested viruses were also detected in nine newly planted symptomless seedlings from the MS-2-F locality. The growers at the examined sites use their own plant propagation material to some extent; strawberry plants are grown in the same plot (or in the close vicinity of older plantings), and *C. f**ragaefolii* occurrence was recorded at the Moravian-Silesian site. In contrast, at one farm in the Zlín Region (locality Z-2-F) where self-propagated material was also used, neither symptoms (except for mild dwarfism probably caused by the cyclamen mite, *Phytonemus pallidus* (Banks, 1901)) nor any of the tested viruses were recorded (Appendix A). At another site in the Zlín Region (locality Z-1-F) where *C. f**ragaefolii* aphids were frequently found and the seedlings originated from the Netherlands, no SPV1 occurrence was recorded among 29 examined plants, and only a few plants were SMoV, SCV, or SMYEV positive. An increase in SPV1 positivity was recorded in production fields in South Bohemian Region (locality SB-2-F, Appendix A). There was only one SPV1-positive plant of *F. ananassa* cv. Darselect among 8 plants screened there in 2019, but there were already 9 positive plants among 17 tested in 2020.

Altogether, SPV1 was detected in a breeding nursery (Liberec Region) and in 16 out of 25 production strawberry farms. The only regions where SPV1 was not detected were Olomouc and Zlín (Figure 5).

### 3.3. Putative Vectors and Non-Strawberry Hosts of SPV1

None of the 19 non-strawberry plant species growing in close vicinity to the sampling sites were identified as a natural SPV1 host (Appendix B). On the contrary, SPV1 was detected in several strawberry-associated aphid species (e.g., *Aphis forbesi* (*A. forbesi*) (Weed 1889), *A. gossypii, A. ruborum, A. sanguisorbae, A. solani, Myzus ascalonicus* (*M. ascalonicus*) (Doncaster 1946) and *C. fragaefolii* (Appendix C and Appendix D). Following the SPV1 detection we estimated viral titers in individual aphids of *A. sanguisorbae*, *A. gossypii* and *C. fragaefolii*. Based on the results (Appendix E), melon aphid, *A. gossypii*, was selected as an optimal vector for further transmission experiments.

### 3.4. Aphid-Mediated Transmission

To support the findings of putative aphid vectors we tested aphids’ ability to transmit SPV1. Transmission trials performed using 16 combinations of different AAPs and IAPs with *A. gossypii* and *F. vesca* ‘Alpine’ No. 814 did not result in SPV1 transmission to ‘Alpine’ seedlings under the experimental conditions. All 64 recipient plants were negative for SPV1 by RT-PCR, while 19 of the 64 aphid batches used for inoculation were SPV1 positive. The highest ratio of positive aphids was obtained from the 48 h AAP (13 positive batches out of 16 tested). A period of at least four hours was needed for *A. gossypii* to acquire SPV1 (2 positive batches out of 16 tested, Appendix A). No aphids were SPV1-positive by RT-PCR after a 10 min AAP. On one recipient plant, pronounced mosaic symptoms appeared on newly emerging leaves from 5 dpi onwards. The plant was later shown to be SMoV positive (48 h AAP, 4 h IAP; Appendix A). The other ‘Alpine’ plants remained asymptomatic. Similarly, four hours were needed for *C. fragaefolii* to acquire SPV1 (all four examined aphids´ batches were RT-PCR positive).

Using eight individuals of *C. f**ragaefolii* feeding on *F. vesca* cv. Rujana 7/2017 subjected to an IAP of 1 month, SPV1 and SCV were identified by RT-PCR and Sanger sequencing in the *F. vesca* ‘Alpine’ recipient plant (SPV1: 1679 nt; Acc. No MW387977) as well as in aphid offspring. The ‘Alpine’ plant showed symptoms of epinasty, irregular vein clearing, leaf malformation, and light chlorotic spots on newly growing leaves (Appendix A). RT-PCR testing for other viruses (SMoV, StrV-1, and SMYEV) returned negative results.

Moreover, ‘Alpine’ plants subjected to IAPs of 1, 2, 3, 5, 7, 11, 14 and 17 days with *C. fragaefolii* were found to be positive for SPV1 by RT-PCR. No disease symptoms were observed on eight SPV1-positive ‘Alpine’ plants.

When individual *A. g**ossypii* and *A. sanguisorbae* aphids were used to transfer SPV1 from *F. ananassa* ČRM3 (IAP 24 and 48 h), none of the 60 ‘Alpine’ recipient plants were found to be positive for SPV1 by RT-qPCR at 60 dpi. However, all 60 aphids tested positive for SPV1. The Ct values varied from 18 to 31 (median Ct 27). The internal control (amplification of the COI gene sequence) revealed Ct values of 21 to 27 (median Ct 23), while no amplification of the internal control specific for plant genetic material (ndhB mRNA) was observed.

### 3.5. HTS Sequencing and Variability of SPV1 Isolates

In total, SPV1 isolates from twenty-two strawberry plants, six arthropods, two indicator ‘Alpine’ plants and one honeydew sample were subjected to Sanger sequencing and deposited in GenBank. Furthermore, the nearly complete genomes (complete all CDSs) of three isolates were obtained by de novo assembly of HTS reads (Table 1). The nt and aa sequences of *F. ananassa* SPV1 isolates from different locations were nearly identical (19 isolates shared 99 to 100% identity of the nt sequences of a 1600 nt-long fragment of P1-P2; two whole-genome sequences were 99.5% identical) and differed from those of the isolates of Rujana (the maximal identities of *F. ananassa* isolates to the Rujana sequences were 97% for the 1600 nt-long fragment of the P1-P2 genes and 95% for whole-genome sequences).

The comparison of the aa sequences of individual genes of 3 Czech and 10 isolates with full-genome sequences in GenBank revealed at least two different phylogenetic clusters, while the Rujana isolates of SPV1 showed the greatest distance from all others (Appendix A). This clustering was supported by the nt and aa sequences of a 1600 nt fragment obtained by the Sanger sequencing of RT-PCR products (Appendix A).

Differences were found between the Rujana isolate and other SPV1 isolates, especially in the P1 and P5 genes. In the P1 gene, the Rujana isolate differed in 99 nts from the reference sequence of SPV1 (NC_025435); in the P5 gene, 90 nts were different. Nevertheless, most of these mutations were synonymous (67 in the P1 gene and 71 in the P5 gene) and did not change the encoded aa residues (Appendix A).

### 3.6. Recombination between SPV1 Genomes

The analysis of nearly full-genome sequences revealed that some parts of four American SPV1 isolates (MZ351169, MZ351170, MZ351171, and MK142237) were highly similar to that of the Rujana isolate (MW387995), while the remainder of their genomes differed substantially. In particular, the MZ351169 (USA) and MZ351171 (USA) sequences shared high similarity to the 5’-end of the P1-P2 gene of the Rujana isolate, and the MK142237 (Argentina) and MZ351170 (USA) sequences showed similarity to the 3’-end of the P5 gene of the Rujana isolate. Detailed analysis with RDP5 software revealed potential recombination points (Table 2) notably in the P1 and P5 genes (Figure 6 and Appendix A). These were further supported by phylogenetic trees of recombined and non-recombined parts of P1 and P5 genes, respectively (Figure 6 and Appendix A).

## 4. Discussion

There is limited knowledge about SPV1 occurrence worldwide. It was noted that estimation of biological significance is the key challenge that follows discovery of a new virus species [21]. In the current study, we focused on characterization of SPV1 occurrence, tested non-strawberry plants from close vicinity to sampling sites, then determined and experimentally verified its putative vectors.

SPV1 was first identified in strawberry samples from Canada by HTS. Due to its high incidence in strawberry plants on both American continents [4,12,13,14] and from the present study in the Czech Republic (Central Europe) (35% positive among 296 examined *F. ananassa* plants; 75% positive among 16 examined *F. vesca* cv. Rujana plants), it can be assumed that SPV1 is present in production plantings in other countries in Europe and worldwide.

As previously described, strawberry viruses commonly occur in mixed infections [2,11,12,22]. Our survey confirmed this, as SPV1 was usually found in coinfections with other examined viruses (77%, i.e., 89 plants out of 115). Nevertheless, we were not able to establish an association between the type of symptoms and the presence of the five tested viruses. Prominent premature leaf reddening and death of strawberry plants at sites in South Moravian, Olomouc and Zlín regions were not usually associated with the presence of viruses, but fungi were identified at sites in Zlín Region. Previously, *Phytphthora fragariae* var. *fragariae*, *Verticillium* sp. and *Rhizoctonia fragariae* were described as associated with strawberry collapsing and severe damage [1]. As we detected *Verticilium* sp. and *Rhizoctonia* sp. in examined strawberries in Zlín Region, the symptoms of leaf reddening and plant death can be fungi-associated. On the other hand, the weakening of strawberry plant growth and premature leaf reddening and leaf decline on the farm in the Pilsen Region was related to the presence of viruses. Among 39 of these plants, only 2 were negative for all of the tested viruses, while 31 of them were positive for SPV1. However, the cause of premature dieback, especially in older plants, needs to be investigated comprehensively. For example, 2- to 3-year-old virus-positive plants of cv. Elkat often showed typical symptoms of viral disease, such as mosaic and irregular vein clearing. Similar symptoms have already been described in this cultivar during a StrV-1 prevalence survey of StrV-1-infected plants [7]. Interestingly, nine young cv. Elkat plants infected with various combinations of the tested viruses did not display any disease symptoms. Other cultivars showed rather nonspecific symptoms, such as leaf stunting, chlorosis, and curling.

The practices followed in a private garden in Třísov (SB-10-G) provide a typical example of the repeated cultivation of strawberries by hobby gardeners. Although the strawberries planted at this site were grown from seeds and were thus presumably virus-free, within one year, their further cultivation was terminated specifically because of virus infection. This was probably due to the simultaneous cultivation of *F. vesca semperflorens*, which seems to be a more attractive plant to aphids than *F. ananassa*. The other important factor is the presence of virus vectors, including other invertebrates (leafhoppers).

Detection of SPV1 amplicons by RT-PCR revealed possible SPV1 vectors and contributed to the understanding of the SVP1 cycle in nature. During the survey, we detected SPV1 sequences in all aphids colonizing SPV1-positive strawberries (oligophagous aphids: *A. forbesi*, *A. ruborum*, *A. sanguisorbae*, and *C. f**ragaefolii*; polyphagous aphids: *A. gossypii*, *A. solani*, and *M. ascalonicus*). We also found SPV1 sequences in honeydew produced by aphid colonies and in the bodies of ants. Honeydew serves as a food for ants, which probably explains why we detected SPV1 in ant bodies. However, ants can protect aphids against natural enemies and transport them to the most susceptible sites on individual plants, which may indirectly contribute to the spread of the virus from plant to plant. Leafhoppers are more commonly associated with phytoplasma diseases, and their role in SPV1 transmission remains unknown. The RT-PCR detection of SPV1 in the larva of the dirt-colored bug of the *Rhyparochromidae* family and in *F. galba* enchytraeids may be associated with feeding on diseased plants, but the role of these species in SPV1 transmission deserves further attention in future. Until recently, poleroviruses were thought to be transmitted solely by aphids. However, pepper whitefly-borne vein yellows virus was recently proven to be transmitted by the whitefly *Bemisia tabaci* [23]. Absence of SPV1 in plants other than *Fragaria* growing near strawberry fields might indicate a limited SPV1 host range, but it requires more attention in future as in the current study, numbers of tested plants for distinct species were quite limited. We may speculate that due to their perennial life cycle, *Fragaria* sp. may serve as the main SPV1 reservoir, if not the only one.

After the successful establishment of *A. gossypii* and *C. fragaefolii.* aphid colonies, we verified the hypothesis that these species can serve as natural vectors for SPV1. Although *A. gossypii* showed the highest SPV1 levels, transmission by single aphids with limited IAP (up to 48 h) under the tested experimental conditions was not successful. Then, we performed these tests with *C. f**ragaefolii*, a known strawberry virus vector [10], and increased both the number of aphids per transmission and the IAP. When eight or more individuals (*C. f**ragaefolii*) were used with an unrestricted IAP, all recipient *F. vesca* ‘Alpine’ plants were infected. Successful SPV1 transmission was achieved by using 10-aphid batches with IAPs varying from 1 to 17 days. SPV1 was transmitted when both *F. vesca semperflorens* infected with multiple virus infection (Rujana 7/2017) as well as *F. vesca* ’Alpine’ infected solely with SPV1 were used as source of inoculum. As an IAP of at least one day was needed, SPV1 was shown to be transmitted in a persistent manner, similar to other poleroviruses [24].

None of the eight SPV1-infected ‘Alpine’ plants showed any symptoms. Therefore, we assume that asymptomatic SPV1 may exist in ‘Alpine’ indicator plants. The symptoms of leaf curl and irregular vein clearing observed in SPV1- and SCV-positive ‘Alpine’ plants were likely due to mixed infections. Similar symptoms have been previously described as characteristic of mixed infections involving SVBV and latent A crinkle virus [10]. Therefore, molecular methods are indispensable for SPV1 detection.

During the survey, we analyzed the variability of the nucleotide sequences of the Czech SPV1 isolates. Interestingly, the *F. ananassa* isolates from distant localities showed nearly identical RdRP gene sequences, suggesting the potential spread of SPV1 via propagation material. Substantially different nucleotide sequences were identified in isolates from *F. vesca semperflorens* cv. Rujana and the associated arthropods at one locality in South Bohemian region (Třísov, SB-10-G). Nevertheless, most of the identified nucleotide differences were synonymous (usually located at the 3rd position of the codon) and did not change the encoded amino acid. Detailed nucleotide comparisons revealed that the sources of SPV1 variation included not only single-nucleotide mutations but also the recombination of larger genomic fragments, similar to what has been reported previously for solemoviruses [25,26]. The majority of the observed nucleotide variability in the Rujana SPV1 isolate was identified in the P1 and P5 genes. As the P5 protein is related to the insect-mediated transmissibility of luteovirids [27,28,29], the preservation of its aa sequence suggests the existence of selection pressure on transmission by the involved insect species. Further nonsynonymous mutations or recombination in this region may, however, influence the transmissibility of the virus and/or its vector range in the future [29,30]. In the present study, one of the identified recombined fragments showed high similarity between the Rujana isolate and two American SPV1 isolates (H2429, H2400), and the second showed similarity to two other American isolates (15CA, H2470). The presence of these two fragments in one genome from the Rujana isolate, together with the other dissimilarities between the Rujana isolate and other sequences, may indirectly indicate that an isolate closely related to the Rujana isolate was a possible source from which these fragments recombined. The fact that the isolates potentially participating in recombination (or their descendants) are from geographically distant locations (America and Europe) emphasizes the need for further analyses of the host range of SPV1 and its vectors to not only understand its molecular biology but to also effectively reduce potential damage caused by the virus or new variants thereof.

## 5. Conclusions

Ultimately, SPV1 was the most common virus in our survey, although it was not detected in all areas. Two-thirds of all identified cases involved coinfection with other strawberry viruses. Neither SPV1 nor any of other tested viruses alone was significantly associated with symptomatic phenotype. Interestingly, experimentally SPV1-infected *F. vesca* ‘Alpine’ strawberry indicator plants did not show any disease symptoms. Recombination events were documented within SPV1 genomes for the first time; however, their impact on biological traits is to be determined.

*C. fragaefolii* aphids were experimentally verified as an SPV1 vector with a persistent manner of transmission, requiring a minimal inoculation access period of 1 day. Successful SPV1 transmission was achieved when using at least eight aphids. Other aphid species (*A. forbesi*, *A. gossypii, A. ruborum*, *A. sanguisorbae*, *A. solani* and *M. ascalonicus*) were also SPV1-positive and are highly likely to contribute to virus transmission. Before consideration on inclusion of SPV1 into phytosanitary diagnostic protocols, given its high incidence rates it is necessary to verify impact of SPV1 infection on different strawberry varieties.

## Figures and Tables

**Figure 1 viruses-13-02487-f001:**
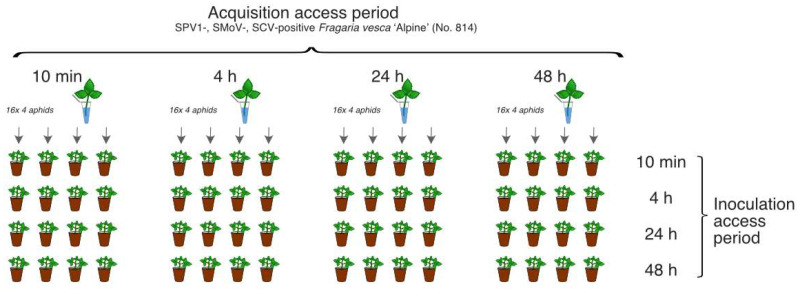
Scheme of the SPV1 transmission experiment using *Aphis gossypii*. SPV1—strawberry polerovirus 1, SMoV—strawberry mottle virus, SCV—strawberry crinkle virus.

**Figure 2 viruses-13-02487-f002:**
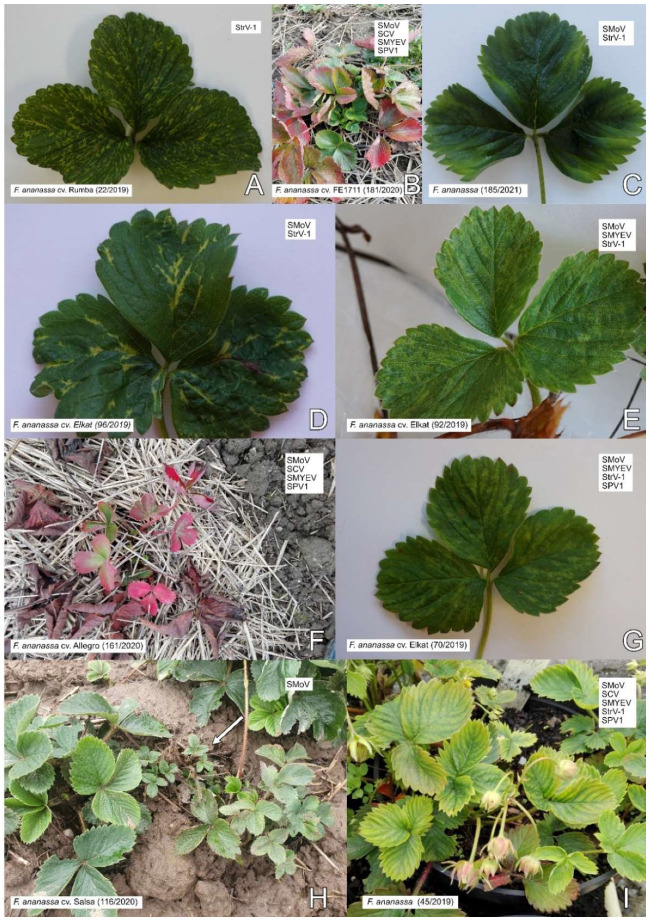
Virus-like disease symptoms on strawberries: (**A**,**D**) irregular vein clearing; (**B**) premature reddening of older leaves and chlorosis of young leaves; (**C**) chlorosis of veins at the leaf margin; (**E**,**G**) mosaic; (**F**) premature dieback and reddening of older leaves together with weakening of the plant; (**H**) stunting (arrow); (**I**) chlorosis of the whole plant. The identification of plants, cultivars, and viruses detected by RT-PCR is indicated in the figure. SMYEV—strawberry mild yellow edge, StrV1—strawberry virus 1.

**Figure 3 viruses-13-02487-f003:**
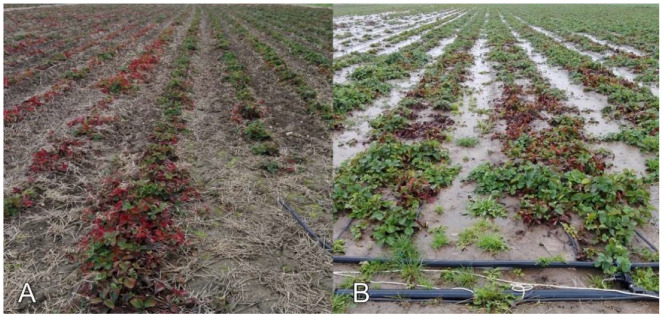
Fields with strawberry plants affected by decline syndrome at P-1-F and Z-1-F sites in Pilsen (**A**) and the Zlín (**B**) regions.

**Figure 4 viruses-13-02487-f004:**
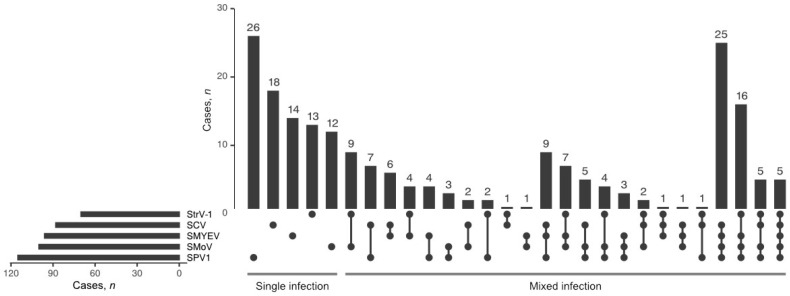
An Upset plot of the coinfection of 332 plant samples by SMoV, SCV, SMYEV, StrV-1, and SPV1.

**Figure 5 viruses-13-02487-f005:**
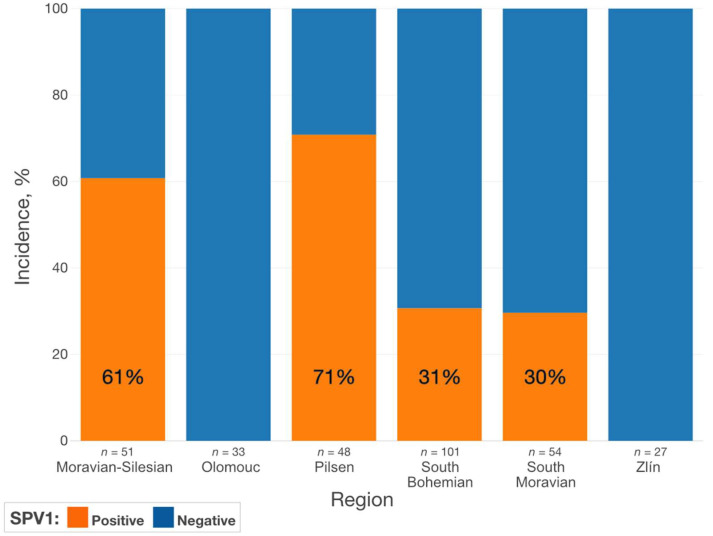
Prevalence of SPV1 infection by a region.

**Figure 6 viruses-13-02487-f006:**
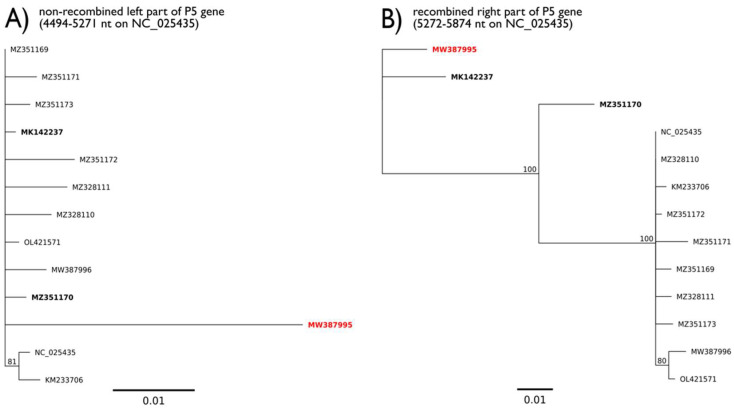
Phylogenetic comparison of non-recombinant (**A**) and putative recombinant (**B**) parts of the P5 nucleotide sequences for the 5276 nt breakpoint. The gene was divided according to the recombination points obtained by RDP5 program (the breaking point on 5276 nt of alignment). Potential recombinants are in bold, the Rujana SPV1 isolate (the Czech recombinant) is in red, nucleotide positions are according to the NC_025435 reference.

**Table 1 viruses-13-02487-t001:** Comparison of SPV1 sequenced with GenBank isolates (nucleotide BLAST).

Sample				Related SPV1 Sequence		
Isolate	Plant/Arthropod Tested	Part of Genome	GenBank	% Identity		
			Acc. No.	(Length)	Acc. No.	Origin
Prague Region						
CB315	*F. ananassa* cv Darselect	P1-P2 gene	MW387989	99.4 (1696)	KM233706	Canada
Central Bohemian Region						
CB191	*F. ananassa* cv Darselect	P1-P2 gene	MW387986	99.4 (1691)	KM233706	Canada
South Bohemian Region						
35/2017	*F. ananassa* cv Darselect	P1-P2 gene	MW387980	99.5 (1674)	KM233706	Canada
23/2020	*F. ananassa* cv Darselect	P1-P2 gene	MW387994	99.5 (1696)	KM233706	Canada
34/2016	*F. ananassa*	complete CDSs	MW387996	99.1 (5947)	MZ328110	Canada
1/2017	*F. vesca semperflorens* cv Rujana	complete CDSs	MW387995	96.0 (5948)	MK142237	Argentina
1/2017-Ho	*Aphis ruborum*’s honeydew	P1-P2 gene	MW387973	97.3(1691)	MZ351170	USA
1/2017-Ar	*Aphis ruborum*	P1-P2 gene	MW387976	97.3 (1693)	MZ351170	USA
1/2017-Ma	*Myzus ascalonicus*	P1-P2 gene	MW387974	96.9 (2416)	MZ351170	USA
1/2017-Rh	*Rhyparochromidae* sp.	P1-P2 gene	MW387975	97.4 (1211)	MZ351170	USA
7/2017	*F. vesca semperflorens* cv Rujana	P5-gene	OL421566	96.5 (1439)	MK142237	Argentina
7/2017-FvA2	*F. vesca* cv Alpine	P1-P2 gene	MW387977	97.3 (1679)	MZ351170	USA
7/2017-FvA2	*F. vesca* cv Alpine	P5-gene	OL421567	96.5 (1439)	MK142237	Argentina
814-Ag_444	*Aphis gossypii*	P5-gene	OL421569	99.2 (1424)	KM233705	Canada
814-Ag_505	*Aphis gossypii*	P5-gene	OL421570	99.2 (1439)	KM233705	Canada
814-FvA	*F. vesca* cv Alpine	P5-gene	OL421568	99.2 (1439)	KM233705	Canada
Pilsen Region						
116/2017	*F. ananassa* cv Christine	P1-P2 gene	MW387981	99.4 (1666)	KM233706	Canada
160/2020	*F. ananassa* cv Allegro	P1-P2 gene	OL421564	99.3 (1696)	KM233706	Canada
169/2020	*F. ananassa* cv Laetitia	P1-P2 gene	OL421565	99.2(1696)	KM233706	Canada
132/2017	*F. ananassa* cv Darselect	P1-P2 gene	MW387982	99.1 (1627)	KM233706	Canada
136/2017	*F. ananassa* cv Darselect	P1-P2 gene	MW387983	99.4 (1669)	KM233706	Canada
Ústí nad Labem Region						
CB129	*F. ananassa* cv Sonata	P1-P2 gene	MW387985	99.4 (1689)	KM233706	Canada
*Liberec Region*						
T22/2016	*F. ananassa*	P1-P2 gene	MW387978	99.2 (1693)	KM233706	Canada
T22/2016-Af	*Aphis forbesi*	P1-P2 gene	MW387979	99.1 (1686)	KM233706	Canada
Hradec Králové Region						
CB256	*F. ananassa* cv Darselect	P1-P2 gene	MW387988	99.4 (1694)	KM233706	Canada
Vysočina Region						
CB231	*F. ananassa* cv Darselect	P1-P2 gene	MW387987	99.5 (2370)	KM233706	Canada
South Moravian Region						
185/2017	*F. ananassa* cv Darselect	P1-P2 gene	MW387984	99.4 (1601)	KM233706	Canada
9/2019	*F. ananassa* cv Symphony	P1-P2 gene	MW387990	99.1 (1696)	KM233706	Canada
100/2019	*F. ananassa* cv Symphony	P1-P2 gene	MW387993	99.1 (1696)	KM233706	Canada
Moravian-Silesian Region						
85/2019	*F. ananassa* cv Elkat	P1-P2 gene	MW387992	99.3 (1692)	KM233706	Canada
66/2019	*F. ananassa* cv Faith	P1-P2 gene	MW387991	99.3 (1644)	KM233706	Canada
138/2020	*F. ananassa* cv Elkat	complete CDSs	OL421571	99.1 (5985)	MZ328110	Canada

**Table 2 viruses-13-02487-t002:** Recombination events detected by RDR5 program between SPV1 isolates. Only recombination events supported by five or more methods were considered; sequences analyzed in detail are in bold. Methods: R = RDP5, G = GeneConv, B = Bootscan, M = MaxChi, C = Chimaera, S = SiScan, T = Topal.

Sequence	Recombination Event Detected				Average *p*-Value	Detection Results						
(acc. num.)	Region 1		Region 2		(RDP)	R	G	B	M	C	S	T
	nt	Gene(s)	nt	Gene								
**MK142237**	1–658	P0, P1	5276–5990	P5	1.24 × 10^−14^	+	+	+	+	+	+	+
**MZ351170**	1–1129	P0, P1	5579–5990	P5	2.46 × 10^−2^	+	-	+	+	+	+	+
**MZ351171**	889–1660	P1	-	-	1.15 × 10^−2^	+	+	+	+	+	+	+
**MZ351169**	1298–1660	P1	-	-	5.14 × 10^−3^	+	+	+	+	+	+	+
MZ351173	1–1624	P0, P1	5286–5990	P5	1.64 × 10^−4^	+	+	+	+	+	+	+
MZ328111	1–888	P0, P1	5949–5990	-	1.66 × 10^−3^	+	+	+	+	+	+	+
MZ328111	3146–3848	P1–2	-	-	3.22 × 10^−9^	+	+	+	+	+	+	+
MZ351169	1661–2451	P1–2	-	-	1.74 × 10^−5^	+	+	+	+	+	+	+
MZ351171	1661–2656	P1–2	-	-	3.41 × 10^−6^	+	+	+	+	+	+	+

## Data Availability

The genomic sequences obtained in this study were submitted to the GenBank database under accession numbers MW387973–MW387997, OK181865, OK181896, OK181901, OK181906, OK181931, OK205264, OK275083, and OL421564–OL421571.

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
