# Peer review of "High Incidence of Strawberry Polerovirus 1 in the Czech Republic and Its Vectors, Genetic Variability and Recombination"

_viruses, 2021, doi:10.3390/v13122487_

Round 1

Reviewer 1 Report

In the current study, strawberry plantations in the Czech Republic as well as wild specimens were tested against the newly discovered strawberry polerovirus SPV1 and five other viruses. Virus presence in potential vectors was examined and transmission trials with aphid vectors were performed. The study is very extended and the analyses well performed providing reliable results concerning the virus incidence and genetic variability while offering new data on epidemiological aspects. My only considerations regard the presentation of the results, in order to be comprehensible to a wider spectrum of scientific public.

Specific comments:

-l. 43: I would suggest to add here the aims of this study instead of only listing out what analysis you have performed

-l. 46: Please offer some more taxonomic information about the other viruses you have selected to test and the reasons why these ones were selected.

-l. 66: Why invertebrates? Have you tested potential vectors other than arthropods? In this study, I have only read about arthropods, please correct me if I missed some information.

-l. 110: Please keep the normal shortened latin name A. gossyppi und A. sanguisorbae.

-I suggest to merge sections 2.3, 2.4 and 2.5 regarding treatment of nucleic acids from plant material together.

-l. 181: I guess the numbering “1.1. Symptoms” is wrong

- l. 154: In this trial, I have understood that the IAPs were extended (3,5,7,11,14,17 days) but not indefinite. The AAP is not concretely given, but it for sure not indefinite. Could you please be more explanatory on defining these periods?

-l. 228-230: Please could you explain the meaning of this finding?

-l. 258: Providing a map would facilitate the understanding. The way farms are named (like P-1-F) are really hard for the reader to follow. Maybe just call farms A, B, C, forest site D and so on?

-l. 414: In the results section, I am missing the output of the HTS analysis.

-l. 428 “fungi were identified at sites in Zlín Region”. How is the sentence associated with the previous text? Maybe the symptom development was correlated with fungal infections? Which fungi do you mean? Please be more concrete.

-l. 446 “including aphids and other invertebrates”: You have already mentioned aphids in the previous sentence.

-l. 447 “Aphids are known vectors of strawberry viruses”: After reading this paragraph I understood that its subject is not the importance of aphids in virus transmission, but rather discussing your RT-PCR results in arthropods and other substances eg honeydew. I would suggest to change the introduction-sentence.

-l. 482 “at locality SB-10-G”: Please use the location name instead, it is hard for the reader to resume this information.

- In the discussion and Conclusions section, I am missing bridging of your results with ones of the international scientific community, specifically as far as biosecurity and phytosanitary measures are concerned. Proposed literature could be from Massart et al. 2017 DOI: 10.3389/fmicb.2017.00045 or Hou, W.; Li, S.; Massart, S. Is There a “Biological Desert” With the Discovery of New Plant Viruses? A Retrospective Analysis forNew Fruit Tree Viruses.Front. Microbiol.2020,11, 592816.

Author Response

Dear Reviewer, 

we appreciate your constructive criticism and valuable suggestions. Below we responded to the question point-by-point. 

In the current study, strawberry plantations in the Czech Republic as well as wild specimens were tested against the newly discovered strawberry polerovirus SPV1 and five other viruses. Virus presence in potential vectors was examined and transmission trials with aphid vectors were performed. The study is very extended and the analyses well performed providing reliable results concerning the virus incidence and genetic variability while offering new data on epidemiological aspects. My only considerations regard the presentation of the results, in order to be comprehensible to a wider spectrum of scientific public. 

Specific comments: 

Q: -l. 43: I would suggest to add here the aims of this study instead of only listing out what analysis you have performed 

A: The relevant parts were modified accordingly. 

Q: -l. 46: Please offer some more taxonomic information about the other viruses you have selected to test and the reasons why these ones were selected. 

A: the text was updated 

Q: -l. 66: Why invertebrates? Have you tested potential vectors other than arthropods? In this study, I have only read about arthropods, please correct me if I missed some information.  

A: All insects examined are members of phylum Arthropoda, but we examined also enchytraeid Fridericia galba, which is member of phylum Annelida. That is why we use “invertebrates” and no “arthropods” 

Q: -l. 110: Please keep the normal shortened latin name A. gossyppi und A. Sanguisorbae. 

A: - done 

Q: -I suggest to merge sections 2.3, 2.4 and 2.5 regarding treatment of nucleic acids from plant material together. 

A: corrected  

Q: -l. 181: I guess the numbering “1.1. Symptoms” is wrong 

A: corrected 

Q: - l. 154: In this trial, I have understood that the IAPs were extended (3,5,7,11,14,17 days) but not indefinite. The AAP is not concretely given, but it for sure not indefinite. Could you please be more explanatory on defining these periods? 

A:  With extended IAPs we used aphids that were raised on SPV1-positive plants, meaning that they unrestricted access to the virus-positive tissues. The wording of that part was changed to clarify it. 

Q: -l. 228-230: Please could you explain the meaning of this finding? 

A: No significant correlation was established between presence/absence of symptoms and presence of any of the detected viruses. In other words, presence of any of viruses was not associated with symptomatic phenotype (i.e. symptoms vs SPV1; symptoms vs SMoV; and so on). The sentence was modified to acknowledge the fact that only F.ananassa samples were included into the analysis. To avoid a wrong assumption that viral infection does not influence plant status, we conducted another analysis, where we evaluated status of the plant appearance (symptomatic/symptomless) and virus infection (infected/uninfected, regardless of details: single or mixed). That showed that symptomatic phenotype was significantly correlated with virus infection.  

Q: -l. 258: Providing a map would facilitate the understanding. The way farms are named (like P-1-F) are really hard for the reader to follow. Maybe just call farms A, B, C, forest site D and so on? 

A: Map has been added as Figure S1. The designation of localities (where the sampling was carried out) in the manuscript is based on the region (SB: South Bohemia Region, P: Pilsen Region, etc.), followed by the locality number and a letter indicating strawberry cultivation either in a production field (F), in a nursery (N), in a hobby garden (G) or a finding in a wood (W).  

Q: -l. 414: In the results section, I am missing the output of the HTS analysis. 

A: the main result of the HTS was reconstitution of the SPV1 sequences, we assumed that inclusion of the full HTS results will make the text less comprehensive 

Q: -l. 428 “fungi were identified at sites in Zlín Region”. How is the sentence associated with the previous text? Maybe the symptom development was correlated with fungal infections? Which fungi do you mean? Please be more concrete. 

A: the text was updated to include that information 

Q: -l. 446 “including aphids and other invertebrates”: You have already mentioned aphids in the previous sentence. 

A: The word “aphids” was removed from the sentence. 

Q: -l. 447 “Aphids are known vectors of strawberry viruses”: After reading this paragraph I understood that its subject is not the importance of aphids in virus transmission, but rather discussing your RT-PCR results in arthropods and other substances eg honeydew. I would suggest to change the introduction-sentence. 

A: corrected 

Q: -l. 482 “at locality SB-10-G”: Please use the location name instead, it is hard for the reader to resume this information. 

A:  corrected, reformulated 

Q: - In the discussion and Conclusions section, I am missing bridging of your results with ones of the international scientific community, specifically as far as biosecurity and phytosanitary measures are concerned. Proposed literature could be from Massart et al. 2017 DOI: 10.3389/fmicb.2017.00045 or Hou, W.; Li, S.; Massart, S. Is There a “Biological Desert” With the Discovery of New Plant Viruses? A Retrospective Analysis forNew Fruit Tree Viruses.Front. Microbiol.2020,11, 592816.  

A:  the discussion part was updated accordingly to the suggestion 

Reviewer 2 Report

In the submitted manuscript, Franova et al., describe the incidence of Strawberry polerovirus 1 in the Checz republic, while also describing the genetic diversity of this virus at multiple sites, identify incidences of multiple virus infection in strawberries, demonstrate that aphids are capable of vectoring this virus, and examine potential recombination events of this virus. Overall, the paper is well written and clear. The methods are appropriate and well done. I have no major comments. A minor issue is the structure of the conclusion paragraph, which is disjoint, and presented in individual sentences. Further effort should be made to construct a cohesive paragraph for the conclusion section.

Author Response

Dear Reviewer,  

we appreciate your positive response. We responded to your question below. 

Q: A minor issue is the structure of the conclusion paragraph, which is disjoint, and presented in individual sentences.  

Further effort should be made to construct a cohesive paragraph for the conclusion section. 

A: the conclusion part was rearranged 

Reviewer 3 Report

Authors performed a survey to detect strawberry polerovirus 1 (SPV1) in the Czech Republic and found that SPV1 was detected in 35% of tested strawberry samples. Most of them are mixed infections with other known strawberry viruses. The virus was detected in strawberry-associated aphids and aphid honeydew by RT-PCR. Chaetosiphon fragaefolii is a potential vector of SPV1. Czech SPV1 isolates belong to at least two phylogenetic clusters, and at least two recombination events between the ancestor of the Czech isolate and three other isolates from the USA and Argentina. The study was properly conceived and conducted.

Here are my line-by-line comments.

L20: SPV1 was also detected in the honeydew. Does this only apply to Aphis ruborum and Chaetosiphon fragaefolii?

L20-22: This only applies to Chaetosiphon fragaefolii.

L125-126: Temperature range?

L313: cDNA was not from insects (leafhoppers, etc.). Viral RNA from insects was reverse transcribed to cDNA.

L318: Again, cDNA was not from honeydew samples. Viral RNA from the honeydew was reverse transcribed to cDNA.

L504-520: It is weird that each sentence contains only one sentence in this section.

Author Response

Dear Reviewer, 

we appreciate your constructive criticism and valuable suggestions. Please find below our response to your questions point-by-point. 

Authors performed a survey to detect strawberry polerovirus 1 (SPV1) in the Czech Republic and found that SPV1 was detected in 35% of tested strawberry samples. Most of them are mixed infections with other known strawberry viruses. The virus was detected in strawberry-associated aphids and aphid honeydew by RT-PCR. Chaetosiphon fragaefolii is a potential vector of SPV1. Czech SPV1 isolates belong to at least two phylogenetic clusters, and at least two recombination events between the ancestor of the Czech isolate and three other isolates from the USA and Argentina. The study was properly conceived and conducted. 

Here are my line-by-line comments. 

Q: L20: SPV1 was also detected in the honeydew. Does this only apply to Aphis ruborum and Chaetosiphon fragaefolii? 

A: Yes, honeydew produced only by A. ruborum and C. fragaefolii was examined for the presence of SPV1. 

Q: L20-22: This only applies to Chaetosiphon fragaefolii. 

A: The AAP period of 4 hours is valid for both aphid C. fragaefolii and A. gossypii.  The IAP period refers only to C. fragaefolii, since SPV1 experimental transmission by the aphid A. gossypii was not successful. Moreover, additional experiments with C. fragaefolii (finished during review process) showed minimal IAP 1 day. The appropriate changes were made in the whole manuscript (abstract, methods, results, and discussion section). 

Q: L125-126: Temperature range? 

A: added 

Q: L313: cDNA was not from insects (leafhoppers, etc.). Viral RNA from insects was reverse transcribed to cDNA. 

A:  the sentence was changed to avoid wrong assumption 

Q: L318: Again, cDNA was not from honeydew samples. Viral RNA from the honeydew was reverse transcribed to cDNA. 

A:  corrected 

Q: L504-520: It is weird that each sentence contains only one sentence in this section. 

A:  reformulated 

Reviewer 4 Report

This manuscript reports the results of a study well conducted on viruses infecting strawberries in the Czech republic with emphasis on SPV1. The following are my contribution.

The conclusion is missing in the abstract.In section 2.8, I will request the authors to be more specific and state the type of analysis they performed (de novo or reference-based alignment) to guide non-experienced scientists who would like to reproduce the same experiment.

Section 2.9.
The authors did not mention which methods/models (Neighbour-joining, maximum likelihood or the software default setting) were used to draw the trees.
The name of the recombination detection program they used is RDP, not RDP5. 5 refers to the latest version of the software, which is still under improvement. The authors stated that they used the stable version of the software (4.101). If the authors want to keep the number in RDP, they should write RDP4 software, v. 4.101.

Line 215; I did not understand the meaning of (2x) and (1x) after the virus names.

Author Response

Dear Reviewer, 

we appreciate your constructive criticism and valuable suggestions. Please find below our response to your question point-by-point. 

This manuscript reports the results of a study well conducted on viruses infecting strawberries in the Czech republic with emphasis on SPV1. The following are my contribution. 

Q: The conclusion is missing in the abstract. 

 A:  The second half of the abstract summarises all findings, we would prefer to keep conclusions as the end of the manuscript. 

Q: In section 2.8, I will request the authors to be more specific and state the type of analysis they performed (de novo or reference-based alignment) to guide non-experienced scientists who would like to reproduce the same experiment. 

A:  the description of performed steps was added to the 2.9 section 

Q: Section 2.9. 
The authors did not mention which methods/models (Neighbour-joining, maximum likelihood or the software default setting) were used to draw the trees. 

A:  information added to the text  
Q: The name of the recombination detection program they used is RDP, not RDP5. 5 refers to the latest version of the software, which is still under improvement. The authors stated that they used the stable version of the software (4.101). If the authors want to keep the number in RDP, they should write RDP4 software, v. 4.101. 

 A: we corrected the wrong number of the software version in the Methods, RDP5 (Beta version, 5.05) was used for recombination analysis. 

Q: Line 215; I did not understand the meaning of (2x) and (1x) after the virus names. 

 A: these are counts of detected cases, corrected to ‘n = ’